# Case Series of Variable Acute Appendicitis in Children with SARS-CoV-2 Infection

**DOI:** 10.3390/children8121207

**Published:** 2021-12-20

**Authors:** Arnis Engelis, Liene Smane, Jana Pavare, Astra Zviedre, Timurs Zurmutai, Marisa M. Berezovska, Jurijs Bormotovs, Mohit Kakar, Amulya K. Saxena, Aigars Petersons

**Affiliations:** 1Department of Paediatric Surgery, Riga Stradins University, LV-1007 Riga, Latvia; arnis.engelis@rsu.lv (A.E.); astra.zviedre@rsu.lv (A.Z.); zurmutay@gmail.com (T.Z.); aigars.petersons@rsu.lv (A.P.); 2Department of Paediatric Surgery, Children’s Clinical University Hospital, LV-1004 Riga, Latvia; marisamaija@gmail.com; 3Department of Paediatrics, Riga Stradins University, LV-1007 Riga, Latvia; liene.smane@rsu.lv (L.S.); jana.pavare@rsu.lv (J.P.); 4Department of Paediatrics, Children’s Clinical University Hospital, LV-1004 Riga, Latvia; 5Department of Doctoral Studies, Riga Stradins University, LV-1007 Riga, Latvia; jurijs.bormotovs@bkus.lv; 6Department of Anaesthesiology & Intensive Care, Children’s Clinical University Hospital, LV-1004 Riga, Latvia; 7Department of Pediatric Surgery, Chelsea Children’s Hospital, Chelsea and Westminster NHS Fdn Trust, Imperial College London, London W12 0NN, UK; amulyasaxena@gmail.com

**Keywords:** SARS-CoV-2, paediatric abdominal pain, paediatric appendicitis, COVID-19

## Abstract

This case series study consists of six children, aged 5–16 years, admitted to a centralized tertiary paediatric hospital serving a population of 1.9 million with acute appendicitis in the setting of SARS-CoV-2 infection. From the beginning of the pandemic in March 2020 until August 2021, 121 COVID-19-positive children were admitted to the hospital. A total of 49 (40.5%) of these patients presented with gastrointestinal symptoms, of which six were diagnosed with acute appendicitis. Five underwent an appendectomy, while one was treated conservatively. To date, it has been reported that appendicitis may have a plausible association with SARS-CoV-2 infection in children. With COVID-19 cases rising, every medical specialist, including all paediatric surgeons, must be ready to treat common acute diseases with SARS-CoV-2 infection as a comorbidity. Providers should consider testing for this infection in paediatric patients with severe gastrointestinal symptoms. Non-surgical treatment of acute appendicitis in children may gain new importance during and after the COVID-19 pandemic. Further studies are needed to prove the link of causality between COVID-19 and acute appendicitis in children.

## 1. Introduction

As global healthcare systems are dealing with the novel coronavirus disease 2019 (COVID-19) pandemic, more experience has been gained in treating patients with combinations of previously known diseases and COVID-19. Gastrointestinal (GI) manifestations, such as vomiting, diarrhoea, and abdominal pain, are known features of COVID-19 in children and may occur without respiratory symptoms [1,2]. Moreover, systematic literature reviews show a high prevalence of GI symptoms in children with positive Severe Acute Respiratory Syndrome Coronavirus 2 (SARS-CoV-2) infection. Other COVID-19-associated GI manifestations that have been reported include mesenteric lymphadenopathy and ileus [3,4]. To date, it has been reported that appendicitis may have a plausible association with SARS-CoV-2 infection in children. A prolonged viral presence was reported in the GI tract of SARS-CoV-2 patients, while human small intestinal cells can support viral infection and replication [5,6].

Prominent GI symptoms may lead to suspected emergency abdominal disorders, e.g., acute appendicitis. SARS-CoV-2 in children appears to be milder than in adults. Paediatric patients are more likely to remain asymptomatic or develop only mild upper respiratory-tract symptoms. The disease’s severity may be impacted by acute appendicitis as a comorbidity, and potential surgical intervention when required [7,8,9]. Paediatric acute appendicitis (AA) is defined as the patient presenting with right lower-abdominal pain, assessed by a paediatric surgeon based on routine clinical laboratory tests (blood count, serum inflammatory markers), abdominal ultrasonography (USG) and the absence of a definite alternative diagnosis of gastrointestinal disease or a gynaecological cause. AA can be classified as uncomplicated (or simple) and complicated acute appendicitis by clinical severity at presentation. Appendectomy has been a standard treatment in AA since 1735. However, in recent years, conservative therapy of acute appendicitis with antibiotics has become more popular. There is an ongoing debate regarding surgical or intravenous treatment of appendicitis of varying severities within the paediatric population [10].

This report describes a case series of six patients admitted to the Children’s Clinical University Hospital (CCUH) in Riga, Latvia, all of whom were diagnosed with acute appendicitis and who also tested positive for SARS-CoV-2.

## 2. Materials and Methods

This study was designed as a retrospective case study series, and all patients had a positive SARS-CoV-2 infection and AA. Between March 2020 and August 2021, a total of six children aged 5–16 years with SARS-CoV-2 infections and AA were admitted to the CCUH. Diagnoses of acute complicated or uncomplicated appendicitis were established using internal CCUH guidelines, and were treated surgically or conservatively, according to need. Using electronic medical records and demographic characteristics, clinical, investigative, and treatment data were collected. Patients with the main diagnostic criteria of multisystem inflammatory syndrome in children (MIS-C) were excluded from the study [11]. This study was approved by the CCUH. Every patient (along with their parents) who was admitted to the CCUH for treatment signed an agreement (consent form) with the hospital in their first language. This report does not contain any personal information that could lead to any of the patients being identified. 

## 3. Case Presentations

We present a case series of patients with acute appendicitis who also tested positive for SARS-CoV-2. They were admitted to a tertiary paediatric hospital, which is a central medical facility for paediatric surgery and treating COVID-19, serving a population of 1.9 million. Six patients, aged between 5–16 years, admitted with positive COVID-19 tests and gastrointestinal symptoms including acute abdominal pain, were diagnosed with acute appendicitis, and five of them underwent surgery. Four patients had complicated and two had acute uncomplicated appendicitis.

### 3.1. Patient 1

A 15-year-old boy came to the emergency department (ED) with a two-day present-ation of abdominal pain, lack of appetite, and vomiting. On examination, the patient was found to have pain and tenderness on the right side of his abdomen. (Clinical, laboratory, radiological, and intraoperative details for all patients are described in Table 1.) SARS-CoV-2 was detected via the polymerase-chain-reaction (PCR) of his nasopharyngeal (NP) swab. Empiric intravenous (IV) antimicrobial treatment with cefotaxime and metronidazole was started and he was taken to the operating room for a laparoscopic appendectomy.

### 3.2. Patient 2

A 14-year-old boy presented to the ED. He had a 24-h history of nausea, diarrhoea, lack of appetite, and abdominal pain, mostly in the right iliac fossa. SARS-CoV-2 was detected via the PCR of his NP swab. On presentation to the ED, pain and tenderness on the right side of the abdomen were noted by examination. Abdominal ultrasound (US) showed findings consistent with acute complicated appendicitis. Empiric IV antimicrobial treatment with cefotaxime and metronidazole was begun and he was taken to the operating room for a laparoscopic appendectomy. An abdominal fluid culture revealed E. coli. The patient was admitted to the hospital during the first 24 h from the onset of symptoms, but the intraoperative findings of peritonitis and broad intra-abdominal inflammation may indicate that acute COVID-19 infection can speed up the disease course of acute appendicitis.

### 3.3. Patient 3

An otherwise healthy 15-year-old girl presented with a one-day history of generalised abdominal pain, nausea, and vomiting. An abdominal examination found pain and tenderness in the right lower quadrant. An abdominal US showed findings consistent with acute complicated appendicitis. A SARS-CoV-2 nucleic acid test was positive. The patient was initially treated conservatively for acute uncomplicated appendicitis with IV antimicrobial treatment (ampicillin plus metronidazole), but abdominal pain advanced, blood inflammation markers elevated, and therefore treatment was converted to surgery. It is possible this patient already had acute complicated appendicitis on ED admission.

### 3.4. Patient 4

A girl aged 12 years presented with fever, abdominal pain, and painful urination of one-day’s duration. The patient had tested positive for COVID-19 eight days before the onset of abdominal pain. SARS-CoV-2 was detected by the PCR of her NP swab. Per abdominal examination findings revealed superficial and deep tenderness in the right lower abdominal quadrant to palpation and localised tenderness to percussion. An abdominal US showed findings consistent with acute complicated appendicitis. IV antimicrobial treatment with cefotaxime and metronidazole was begun, and she was taken to the operating room for a laparoscopic appendectomy. An abdominal fluid culture revealed P.aeruginosa, Str.viridans, and Gemella morbillorum.

### 3.5. Patient 5

A girl aged 16 years presented with fever, abdominal pain in the epigastric and ileocecal region, nausea, lack of appetite, and vomiting of two days’ duration. Patient 5 had a recurrence of acute uncomplicated appendicitis. She had had the first episode two years previously, with acute uncomplicated appendicitis. She was treated conservatively with antibiotics; however, she was ultimately operated on laparoscopically. In her case, COVID-19 infection presumably exacerbated the course of appendicitis and resulted in abdominal pain that was a cause for diagnostic laparoscopy and further appendectomy. Unlike the four other cases in which the histology showed necrotic areas in the appendix wall, concluding that appendectomy was necessary (gangrenous appendicitis—see Table 1), Patient 5’s surgery could have been avoided if symptoms had not persisted.

### 3.6. Patient 6

A girl aged five years presented with fever, abdominal pain, nausea and vomiting of one day’s duration. She had a recurrence of acute uncomplicated appendicitis. This girl had had her first episode two years previously, when she had acute appendicitis with an appendicular mass. She was treated conservatively with antibiotics; however, Patient 6 once again was treated non-surgically. In her case, the COVID-19 infection presumably exacerbated the course of appendicitis and resulted in abdominal pain.

## 4. Discussion

Infection of SARS-CoV-2 has had a major impact on paediatric surgery since 11 March 2020, when COVID-19 was first officially declared a pandemic [12]. Each patient with suspected surgical disease in an emergency department (ED) must be tested for the SARS-CoV-2 RNA via an oropharyngeal swab or saliva test taking an average 4–6 h [13]. If sample results were not readily accessible and treatment of the disease was urgent, the surgery team had to proceed as if the child had COVID-19 and follow all necessary precautions [14]. 

Acute appendicitis is the most frequent cause of emergency abdominal surgery in children, with a peak incidence in the adolescent age range [15]. Acute uncomplicated appendicitis may be treated solely with antibiotics; however, when it progresses to acute complicated appendicitis, surgical treatment is necessary. Delayed diagnosis and treatment of acute appendicitis may lead to complications such as peritonitis, intra-abdominal abscess, sepsis, ileus, or even patient death [16]. The COVID-19 pandemic gave rise to many fears about hospitalisation and prolonged stays in the ED, considering potential cross-infection and delays in treatment due to disease control measures and other logistical restrictions. These delays impact the course of other diseases such as acute appendicitis, which influences possible adverse outcomes for patients.

Adult COVID-19 patients more commonly present with respiratory symptoms, fever, myalgia, and fatigue. Paediatric patients’ clinical presentations differ, as gastrointestinal (GI) symptoms are more prevalent. Patients may primarily present with nausea, vomiting, diarrhoea, or abdominal pain, without any respiratory symptoms [17,18]. In all six cases reported in this series, patients complained of abdominal pain on ED admission.

Although the development of acute appendicitis may not be a direct sequela of COVID-19, there are reports of possible causality in the current literature [14]. In addition, Meyer and co-authors reported four patients with acute appendicitis and its plausible association with SARS-CoV-2 infection [5].

Thorough examination of COVID-19 patients with gastrointestinal symptoms is necessary. Relying solely on the Alvarado score and clinical presentation may not be sufficient information for the decision to perform surgery. Additional abdominal ultrasound or computed tomography should be performed on COVID-19 patients with suspected acute abdominal surgical diseases. Non-surgical management of acute appendicitis with antibiotics may be the first treatment option for patients with a positive SARS-CoV-2 infection. This step may curtail the course of appendicitis, meaning the patient does not have to undergo surgery and its associated risks. This also relieves the surgical, anaesthestic and intensive care units of the additional workload related to COVID-19 challenges and mandatory precautions.

Another retrospective study of 48 SARS-CoV-2-positive and 61 SARS-CoV-2-negative paediatric patients with acute appendicitis at a tertiary-level children’s hospital found a trend in both groups towards a longer duration of GI symptoms before ED admission in children with appendiceal perforation [19]. This contradicts our experience, as cases Nos. 2 and 4 were admitted to the ED within 24 h of the onset of symptoms and the intraoperative findings were gangrenous appendicitis with perforation. This exaggerated inflammatory reaction leading to appendiceal rupture may be present in SARS-CoV-2 infected patients as a potential characteristic of COVID-19 pathophysiology. Therefore, presumably SARS-CoV-2 can cause cells to express human angiotensin-converting enzyme 2 (ACE2), which induces a large secretion of cytokines which eventually causes a cytokine storm, damaging multiple organs throughout the body [20]. ACE2 is commonly found in human intestinal epithelial cells. He et al. performed autopsies on patients who died of SARS, and the pathological results showed that in ACE2-expressing cells, monocyte chemokine-1 (MCP-1), tumour growth factor-β1 (TGF-β1), tumour necrosis factor-α (TNF-α), interleukin (IL)-1β and IL-6, were highly expressed [21]. Referencing our previous study, serum MCP-1 and IL-6 were significantly increased in children with acute complicated appendicitis [22]. This also could explain the appendiceal rupture within 24 h of disease onset for Patient Nos. 2 and 4. 

## 5. Conclusions

Since COVID-19 cases are rising, every medical specialist, including paediatric surgeons, must be ready to treat common acute diseases with SARS-CoV-2 infection as a comorbidity. Providers should consider testing for this infection in paediatric patients with severe gastrointestinal symptoms, recalling the hypothesis that appendicitis could have a plausible association with SARS-CoV-2 infection. Non-surgical treatment of acute appendicitis in children may become more important during, and after, the COVID-19 pandemic. Further studies are needed to prove the link of causality between COVID-19 and acute appendicitis in children.

## Figures and Tables

**Table 1 children-08-01207-t001:** Clinical characteristics and case definitions of children.

Criteria		Case Nr 1	Case Nr 2	Case Nr 3	Case Nr 4	Case Nr 5	Case Nr 6
Age (Years)		15	14	15	12	16	5
Sex		M	M	F	F	F	F
Diagnosis (includes SARS-CoV-2 Infection in All Patients)		Acute Complicated Appendicitis	Acute Complicated Appendicitis	Acute Complicated Appendicitis	Acute Complicated Appendicitis	Acute Uncomplicated Appendicitis	Acute Uncomplicated Appendicitis
Clinical symptoms	Duration of symptoms (d)	2	1	3	1	2	1
	Fever	No	No	No	Yes	Yes	Yes
	Abdominal pain	Yes	Yes	Yes	Yes	Yes	Yes
	Nausea or vomiting	Yes	Yes	Yes	No	Yes	Yes
	Diarrhoea	No	Yes	No	No	No	No
	Lack of appetite	Yes	Yes	No	No	Yes	No
	Pain migration to RLQ	Yes	Yes	Yes	Yes	Yes	Yes
	Rebound tenderness in RLQ	Yes	Yes	Yes	Yes	No	No
	Alvarado score	8	9	7	7	7	7
Inflammation	CRP, (0–5) mg/L	13.4	25.25	65.17	2.57	0.4	7.58
	IL-6, (0–2) pg/mL	Not performed	13.8	67.2	243	6.28	23.2
	White blood count, (3.84–9.84) ×10^3^/mkl	9.38	10.17	15.65	14.84	10.17	9.8
	Neutrophil, (1.54–7.04) ×10^3^/mkl	8.05	9.12	13.89	11.80	5.65	7.58
	Lymphocyte, (0.97–4.28) ×10^3^/mkl	N/A	0.65	0.94	1.35	3.26	1.25
	Platelets, (175–369) ×10^3^/mkl	165	188	281	240	262	236
	Abdominal culture	Negative	Positive (E.Coli)	Positive (E.Coli)	Positive (P.Aeruginosa, Str.Viridans, Gemella Morbillorum)	Not performed	N/A
Diagnostic approach of SARS-CoV-2 infection	RT-PCR	Positive	Positive	Not performed	Positive	Positive	Positive
	Anti-SARS-CoV-2 total antibodies	Not performed	Not performed	Not performed	Positive	Not performed	Not performed
	SARS-CoV-2 & Influenza A/B Nucleic acid test	SARS-CoV-2—positive; Influenza A/B—negative	SARS-CoV-2—positive; Influenza A/B—negative	SARS-CoV-2—positive; Influenza A/B—negative	Not performed	Not performed	Not performed
	Prior four week exposure	Unknown	Unknown	Unknown	Positive	Unknown	Unknown
	Vaccination status	Unvaccinated	Unvaccinated	Unvaccinated	Unvaccinated	Unvaccinated	Unvaccinated
	MIS-C status	Negative	Negative	Negative	Negative	Negative	Negative
Surgery	Access	Laparoscopic appendectomy	Laparoscopic appendectomy	Laparoscopic appendectomy	Laparoscopic appendectomy	Laparoscopic appendectomy	Non-surgical treatment
	Intraoperative finding	Appendicitis with microperforation	Appendicitis with macroperforation	Appendicitis with macroperforation	Appendicitis with macroperforation	Appendicitis without perforation Apoplexy of right ovary	N/A
	Histology	Gangrenous appendicitis	Gangrenous appendicitis	Gangrenous appendicitis	Gangrenous appendicitis	Appendicitis with lymphocyte infiltrates and fibrosis	N/A
Imaging	Abdominal US	Not performed	US signs of complicated appendicitis	US signs of complicated appendicitis	US signs of complicated appendicitis	Not performed	US signs of uncomplicated appendicitis Mesenteric lymphadenitis
Management	PICU admission	No	No	No	No	No	No
	Antibiotics	Yes	Yes	Yes	Yes	Yes	Yes
	Hospital stay duration, d	5	7	9	11	4	5

No, number; CRP, C-reactive protein; F, female; M, male; MIS-C, multisystem inflammatory syndrome in children; RT-PCR, reverse transcription polymerase chain reaction (PCR) assay; PICU, paediatric intensive care unit; RLQ, right lower quadrant; SARS-CoV-2, severe acute respiratory syndrome coronavirus 2; d, day; US, ultrasound; Il-6, Interleukin 6.

## Data Availability

No new data were created or analysed as part of this study. Data sharing is not applicable to this article.

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
