# Peer review of "Case Series of Variable Acute Appendicitis in Children with SARS-CoV-2 Infection"

_children, 2021, doi:10.3390/children8121207_

Round 1
Reviewer 1 Report
Cases should be presented as description, not as tables. Moreover, some description is found in the discussion, and should be moved to the case presentation section.
Please check the age of the patients in table 1
Please explain clearly the diagnostic workup and the criteria you adopted to decide between surgery or nonoperative treatment
Author Response
Dear Reviewer
Thank you very much for the review of our manuscript “Case series of variable acute appendicitis in children with SARS-CoV-2 infection” (Manuscript-ID 1449963), which we submitted previously to Children. We sincerely appreciate all your valuable feedback, which has helped us to significantly improve the quality of the article. We outline our responses to the reviewers’ comments below, and we have addressed each one in turn. We have made the relevant changes to the manuscript, as you have suggested.
This article has been prepared in accordance with Children manuscript-formatting guidelines. And we stand ready to make any further improvements to the manuscript, as necessary.
I hereby affirm that this manuscript has not been published previously, or accepted for publication, nor is it under consideration for publication elsewhere. All authors have contributed significantly to this work, and have reviewed and approved the final version of the manuscript.
We hope that you will now be able to accept our manuscript for publication in Children.
Yours sincerely
Mohit Kakar, M.D.
Consultant Pediatric surgeon at the Children’s Clinical University Hospital of Latvia
The corresponding author
Responses to the reviewers’ comments
- Reviewer 1:
Cases should be presented as descriptions, not as tables. Moreover, some description is found in the discussion and should be moved to the case presentation section.
Response: Thank you for your accurate remarks. Each case is presented as a description. The most important information about each case has been moved to the case presentation section as you requested. All other data remains in Table 1.
Please check the age of the patients in table 1
Response: Thank you for this comment. On reflection, we can only present general data about our patient population (Table 1), but not information about each individual separately in Table 1, because this research involves only six case studies and not a cohort study. We decided not to leave this table in the article.
Please explain clearly the diagnostic workup and the criteria you adopted to decide between surgery or nonoperative treatment
Response: Thank you for your accurate remarks. Non-surgical treatment of acute appendicitis in children is applied to all uncomplicated cases, while surgical intervention is used only for patients with complicated appendicitis and for those whose appendicitis is so acute non-surgical treatment would not be effective. According to the diagnostic and treatment algorithm of acute appendicitis at Children’s Clinical University Hospital, Riga, the criteria for uncomplicated acute appendicitis are: Alvarado score ≥7, CRP cut-off values 8.4mg/L or IL-6 cut of values 36.2 pg/mL and signs of appendicitis in the US. The criteria for complicated acute appendicitis in the US are Alvarado score ≥7, CRP cut-off values > 8.4mg/L or IL-6 cut-off values > 36.2 pg/mL, signs of appendicitis, and peritoneal irritation symptoms are also presented. Detailed information is explained in the References section No 21 (Zviedre, A.; Engelis, A.; Tretjakovs, P.; Zile, I.; Petersons, A. Laboratory Tests in Addition to the Alvarado Score in the Management of Acute Appendicitis in School-Age Children. Proceedings of The Latvian Academy Of Sciences. 2019, 73, 379–386)
Reviewer 2 Report
This is a retrospective cohort on Corona-positive patients with gastrointestinal complaints which were further diagnosed with appendicitis.
There is no clear research hypothesis given. It remains unclear, if the authors wanted to look into possible changed epidemiology of appendicitis or the unclear relation between COVID and appendicitis.
The authors statement, that during COVID lockdown, the number of complicated appendicitis in children was enlarged is not uniformly proven. There are several articles which do not support this or even show the opposite.
Furthermore, it has been clearly shown, that pediatric patients with pediatric inflammatory multisystem syndrome (PIMS) often present with severe gastrointestinal symptoms. These patients should not be operated.
The authors should work out if the included appendicitis patients truly had COVID or were just SARS-CoV-2-positive. This would make a major difference.
Furthermore, it should be appropriated to have a control group. How many admitted patients with gastrointestinal complaints would have been diagnosed with appendicitis apart from the pandemic, i.e. in a historical group.
How many appendicitis patients have been treated in the observation period not being SARS-CoV-2-positive?
The ratio of complex vs. simple appendicitis is 4:2, which clearly unusual. What is the explanation for that.
The authors statement in the introduction, that advanced stages of appendicitis require surgery are also not completely supported by the literature. There several studies which expressively show that complicated appendicitis could be safely treated with iv. antibiotics.
The here presented study requires clear inclusion and exclusion criteria, a control group, biostatistical planning and an ethical clearance.
Author Response
Dear Reviewer
Thank you very much for the review of our manuscript “Case series of variable acute appendicitis in children with SARS-CoV-2 infection” (Manuscript-ID 1449963), which we submitted previously to Children. We sincerely appreciate all your valuable feedback, which has helped us to significantly improve the quality of the article. We outline our responses to the reviewers’ comments below, and we have addressed each one in turn. We have made the relevant changes to the manuscript, as you have suggested.
This article has been prepared in accordance with Children manuscript-formatting guidelines. And we stand ready to make any further improvements to the manuscript, as necessary.
I hereby affirm that this manuscript has not been published previously, or accepted for publication, nor is it under consideration for publication elsewhere. All authors have contributed significantly to this work, and have reviewed and approved the final version of the manuscript.
We hope that you will now be able to accept our manuscript for publication in Children.
Yours sincerely
Mohit Kakar, M.D.
Consultant Pediatric surgeon at the Children’s Clinical University Hospital of Latvia
The corresponding author
Responses to the reviewers’ comments
- Reviewer 2:
This is a retrospective cohort of Corona-positive patients with gastrointestinal complaints who were further diagnosed with appendicitis.
Response: Thank you for this valuable comment. We are only presenting only six case series study of variable acute appendicitis in children with SARS-CoV-2 infection, with no retrospective cohort study.
There is no clear research hypothesis given. It remains unclear if the authors wanted to look into possible changed epidemiology of appendicitis or the unclear relation between COVID and appendicitis.
Response: Thank you for your accurate remarks. The study hypothesis is that appendicitis may have a plausible association with SARS-CoV-2 infection in children.
The authors' statement, that during COVID lockdown, the number of complicated appendicitis in children was enlarged is not uniformly proven. There are several articles that do not support this or even show the opposite.
Response: Thank you for this comment. The authors’ statement that complicated acute appendicitis in children has grown during the COVID-19 pandemic could be explained not just statistically according to references (Place, R.; Lee, J.; Howell, J. Rate of Paediatric Appendiceal Perforation at a Children’s Hospital during the COVID-19 Pandemic Compared with the Previous Year. JAMA Netw Open. 2020, 3, e2027948) but also with data that SARS-CoV-2 can cause cells to express human angiotensin-converting enzyme 2 (ACE2), which induces a large secretion of cytokines that eventually cause a cytokine storm, damaging multiple organs throughout the body and leading to appendiceal rupture (see the final paragraph of the Discussion section).
Furthermore, it has been clearly shown, that pediatric patients with the pediatric inflammatory multisystem syndrome (PIMS) often present with severe gastrointestinal symptoms. These patients should not be operated.
Response: Thank you for pointing this out. Pediatric inflammatory multisystem syndrome (PIMS) was excluded from this study.
The authors should work out if the included appendicitis patients truly had COVID or were just SARS-CoV-2-positive. This would make a major difference.
Response: Thank you for this comment. All six study patients with acute appendicitis presented with symptomatic SARS-CoV-2 infection.
Furthermore, it should be appropriate to have a control group. How many admitted patients with gastrointestinal complaints would have been diagnosed with appendicitis apart from the pandemic, i.e. in a historical group.
Response: Thank you for this comment. We are presenting only six case series study of variable acute appendicitis in children with SARS-CoV-2 infection, with no retrospective cohort study.
How many appendicitis patients have been treated in the observation period not being SARS-CoV-2-positive?
Response: Thank you for pointing this out. Between March 2020 and August 2021, a total of 190 children with acute appendicitis were admitted to the Children’s Clinical University Hospital in Riga.
The ratio of complex vs. simple appendicitis is 4:2, which clearly unusual. What is the explanation for that?
Response: Thank you for this question. Between March 2020 and August 2021, only six patients were diagnosed with acute appendicitis and symptomatic SARS-
CoV-2 infection. A statistical method was not used to measure the ratio of complex vs. simple appendicitis because it was impossible to do so given that we only had six cases during the study period.
The authors' stated in the introduction, that advanced stages of appendicitis require surgery is also not completely supported by the literature. There are several studies that expressively show that complicated appendicitis could be safely treated with iv. antibiotics.
Response: Thank you for your accurate remarks. The treatment methods for complicated appendicitis are discussed in more detail in the Introduction section as you marked.
The here presented study requires clear inclusion and exclusion criteria, a control group, biostatistical planning, and ethical clearance.
Response: Thank you for your comment. Our study only presents six case studies of variable acute appendicitis in children with SARS-CoV-2 infection, with no retrospective cohort study. Therefore, we do not need any control group or biostatistical planning. Ethical approval of this study was outlined in the Materials and Methods section.
Round 2
Reviewer 1 Report
Ok
Reviewer 2 Report
The authors have responded to the reviewers comments and question in appropriate manner.
The manuscript has improved substantially.